# Prevalence and determinants of insulin resistance in recovered COVID-19 and uninfected residents of two regional capitals in Ghana: An observational study

Ansumana Sandy Bockarie[1], Leonard Derkyi-Kwarteng[2], Jeffrey Amankona Obeng[3], Richard Kujo Adatsi[4], Ebenezer Aniakwaa-Bonsu[5], Charles Apprey[6], Jerry Ampofo-Asiama[7], Samuel Acquah[8]*

**1** Department of Internal Medicine and Therapeutics, School of Medical Sciences, College of Health and Allied Sciences, University of Cape Coast, Cape Coast, Ghana, **2** Department of Pathology, School of Medical Sciences, College of Health and Allied Sciences, University of Cape Coast, Cape Coast, Ghana, **3** Department of Internal Medicine, Cape Coast Teaching Hospital, Cape Coast, Ghana, **4** Public Health Reference Laboratory, Tamale Teaching Hospital, Tamale, Ghana, **5** Department of Microbiology and Immunology, School of Medical Sciences, College of Health and Allied Sciences, University of Cape Coast, Cape Coast, Ghana, **6** Department of Biochemistry and Biotechnology, School of Biosciences, College of Science, Kwame Nkrumah University of Science and Technology, Kumasi, Ghana, **7** Department of Biochemistry, School of Biological Science, College of Agricultural and Natural Sciences, University of Cape Coast, Cape Coast, Ghana, **8** Department of Medical Biochemistry, School of Medical Sciences, College of Health and Allied Sciences, University of Cape Coast, Cape Coast, Ghana

* sacquah@ucc.edu.gh

## Abstract

The long-term impact of the coronavirus disease 2019 (COVID-19) pandemic on metabolic risk factors in different populations has not been fully investigated. Insulin resistance (IR) is a cardinal risk factor for the development of type 2 diabetes mellitus. The current study sought to determine the prevalence and determinants of insulin resistance in selected Ghanaians with and without past COVID-19 status in the Cape Coast and Tamale metropolitan areas. Using a cross-sectional study design involving 510 adult participants, body mass index (BMI), waist-to-hip ratio, systolic blood pressure, lipid profile, insulin, plasma glucose, C-reactive protein (CRP), beta-cell function and insulin resistance levels were measured and compared between participants with and without past COVID-19 status. IR was determined by the homeostatic model (HOMA-IR) and the triglyceride-glucose index (TyG). Percentage prevalence and Poisson regression with prevalence ratio and 95% confidence intervals were applied. IR prevalence ranged from 70.69% to 79.09% (HOMA-IR) and 88.62% to 90.91% (TyG) respectively for Tamale and Cape Coast residents. IR prevalence values of 70.98% and 88% (HOMA-IR) and 89.02% and 90.2% (TyG) for without and with past COVID-19 groups respectively were observed. Irrespective of background, low (31.18%) and high (19.41%) levels of beta-cell function were detected. Additionally, high levels of very-low density lipoprotein cholesterol (8.31%), triglycerides (24.9%),

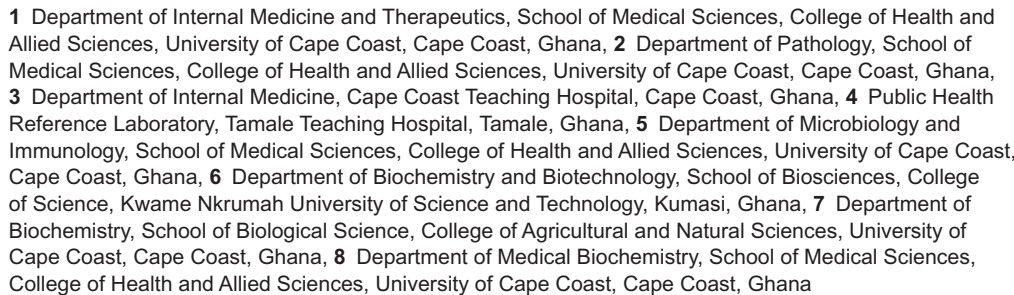

**Data availability statement:** All relevant data are within the paper.

**Competing interests:** The authors have declared that no competing interests exist.

total cholesterol (27.45%), low-density lipoprotein cholesterol (44.71%) and low level of high-density lipoprotein cholesterol (11.96%) coupled with low-grade inflammation (50.59%) were observed. Irrespective of surrogate marker used or past COVID-19 status, age, educational level and triglycerides could significantly associate with IR. With HOMA-IR, fasting plasma glucose, insulin and total cholesterol predicted IR in participants without prior COVID-19 status. With TyG, age, BMI, triglycerides and CRP were the predictors of IR in participants with past COVID-19 status. The risk of development of type 2 diabetes mellitus through insulin resistance is high in our setting. Measures to reduce the rising pace of IR are urgently needed in our setting.

## Introduction

Although the devastation of acute coronavirus disease 2019 (COVID-19) on various populations and world health systems appear to have abated, its long-term effects on the health and wellbeing of the affected is still persisting in various forms. Even in the sub-Saharan African region where the effect of the COVID-19 pandemic was not deemed as severe in terms of morbidity and mortality, the long-term impact on the populace cannot be underestimated. According to the latest data from the Ghana Health Service, the COVID-19 had affected 171,889 since its outbreak in 2020, as at March 2, 2023 with 70.8% of the population vaccinated against the condition [1].

Recent data from the International Diabetes Federation [2] still maintains that the sub-Saran African region has the highest prevalence of undiagnosed cases of diabetes and it is expected to experience the highest incidence of the condition by 2045. This gloomy prediction is hinged on predicted increased trend of obesity, driven by sedentary lifestyle and consumption of a westernized diet dominated by refined carbohydrates and saturated fats [2]. These diabetogenic predictions do not factor the probable role of infection-driven inflammation in the development of diabetes, especially, in the sub-Saharan African region. Indeed, the sub-Saharan African region continues to remain the hub of global infectious diseases [3]. It was probably based on this notion that the region was erroneously predicted to suffer the highest prevalence of mortality associated with acute COVID-19 infection in the early stages of the pandemic. In spite of the relatively mild short-term impact of COVID-19, in terms of morbidity and mortality on the regional populace, the probable long-term effects on the development of chronic non-communicable diseases in the region, especially, type 2 diabetes mellitus (T2DM), remains to be fully ascertained.

Interestingly, some studies have found that COVID-19 in people with diabetes increases diabetic complications and adverse outcomes of the viral condition [4–6]. Other studies [6–8] have associated the COVID-19 condition with new-onset diabetes of type 1 phenotype. Although T2DM constitutes more than 90% of global cases of diabetes, studies on the impact of COVID-19 on the condition post-recovery are limited.

Chen et al. [9] observed that six months after recovery from COVID-19, the risk of insulin resistance could increase. COVID-19-associated insulin resistance is thought

to develop at several points of viral-host interactions and involves the integrated stress response (ISR) system [10]. The ISR includes a series of signaling pathways that are activated in response to various factors including obesity and viral infection and can result in impaired insulin action. Another important biochemical phenomenon that is central to the development of COVID-19-associated insulin resistance is inflammation. Inflammation is a defense mechanism but can be harmful if dysregulated. Inflammation is linked to various health conditions including insulin resistance, diabetes, obesity, cancer and other chronic disease conditions [10,11]. Interestingly, these conditions can also reinforce the inflammatory phenomenon.

Considering the critical role of insulin resistance in the pathogenesis of several chronic diseases in general and T2DM in particular, it is important to examine the promoting factors in our setting in order to identify appropriate mitigating measures to moderate its impact. This is critical because, insulin resistance, which is a cardinal risk factor for development of T2DM, can be influenced by several factors that may vary with the setting, making it difficult to adopt universal mitigating measures without recourse to the specific local context. Therefore, the current study aims to determine the prevalence and determinants of insulin resistance using two surrogate markers in a Ghanaian population that had recovered completely from COVID-19 and their counterparts that never had the condition in the Cape Coast and Tamale metropolitan areas.

## Methods

### Study site, design and population

The study was conducted among residents of Cape Coast and Tamale metropolitan areas with the laboratory measurements taking place at the teaching hospitals of the two cities. The Cape Coast Teaching Hospital (CCTH) and the Tamale Teaching Hospital (TTH) are located in the Cape Coast and Tamale metropolis respectively. Both are tertiary facilities in the Ghanaian health care system and played a critical role in the management of COVID-19 cases during the pandemic. Whereas the CCTH is situated in the Central region along the coast of the Gulf of Guinea, TTH is located in Northern region of Ghana just below the Sahel belt. The two sites differ in geography, climate, ethnicity and dietary norms. The target population was the apparently healthy adults in the two metropolis. This cross-sectional study involved a total of 510 participants with 220 from Cape Coast and 290 from Tamale. The inclusion criteria were adults aged 20 years and above, apparently healthy, vaccinated against COVID-19 and without any underlying health condition at the time of recruitment. The participants who had recovered from COVID-19 had been declared healed from the condition by a negative PCR test results coupled with a clinical determination at least eight months before recruitment and were residents of either Tamale or Cape Coast and their surrounding communities. Those without a history of the COVID-19 condition originated from the same communities as their recovered counterparts but were never diagnosed of the condition. Individuals who were pregnant, breastfeeding, diabetic, or had any health condition that could affect the levels of any of the blood-based measured indices were excluded from the study. The assessment of the inclusion and exclusion criteria was by effective administration of the appropriate questionnaires by trained health professionals. In addition, available health records of recovered COVID-19 were cross-checked with the responses of participants where appropriate. In both towns, workers in the formal sector of the Ghanaian economy dominated the participants. All participants had been vaccinated against the COVID-19 virus. Fig 1 is a sketch of map of Ghana showing Cape Coast and Tamale where the study was conducted.

**Determination of sample size.** The sample size was determined for each study site from the prevalence of COVID-19 at the national and regional levels by assuming the regional prevalence to be specific for the health facility of interest. Therefore, the regional prevalence of COVID-19 for the Central was assumed to be for the CCTH with that of the Northern region for the TTH, due to the critical roles that these tertiary health facilities played in the management of COVID-19 cases in their respective regions. The formula, $N = \{P1(1-P1) + P2(1-P2)\}\left(\frac{Z\alpha + Z\beta}{E}\right)^2$, [12] was applied to determine the sample size for each of the study sites. The estimated regional prevalence (P1) of COVID-19 were 3% and 0.15% for the

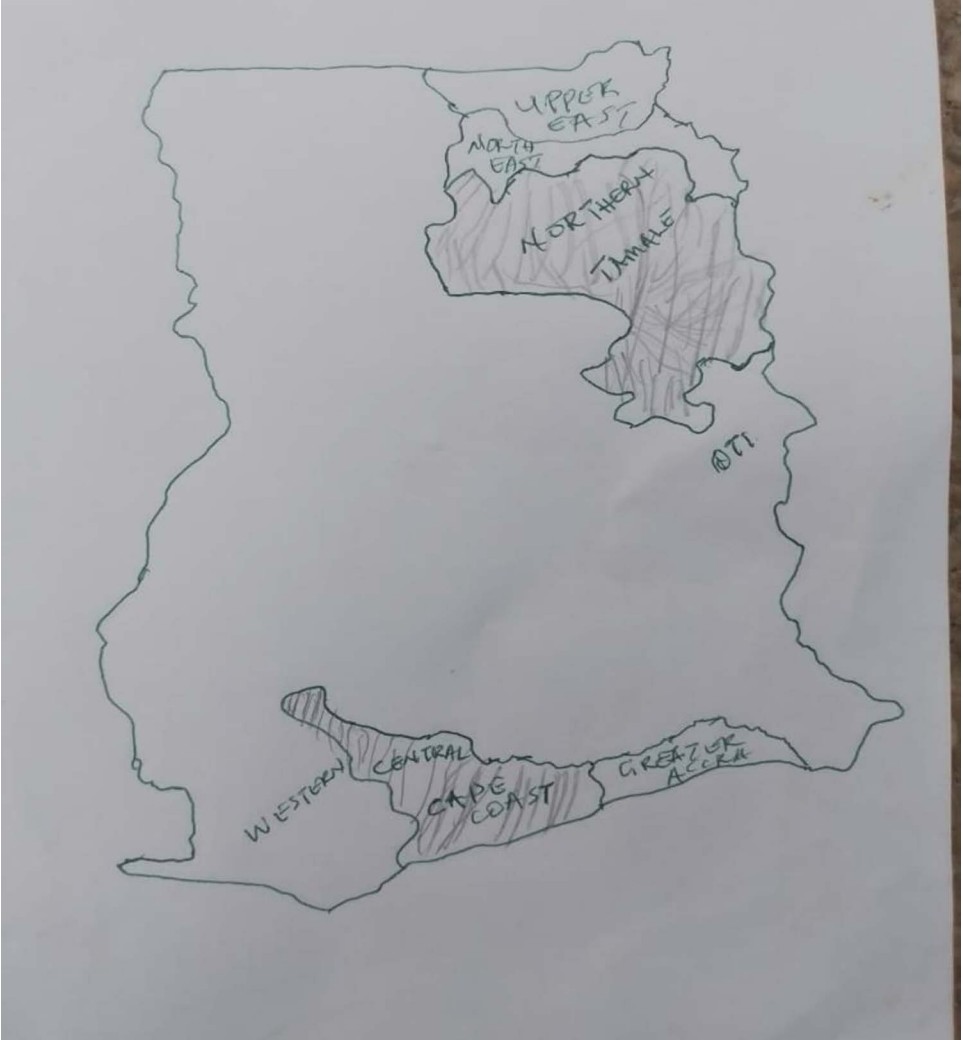

**Fig 1. A sketched map of Ghana showing the regions (shaded where the study was conducted.**

Central and Northern regions respectively coupled with a national prevalence (P2) of 0.5%. However, the Central regional prevalence of the condition was applied across board in estimating the sample size in order to get adequate numbers for the study. Using the 3% regional and 0.5% national prevalence of COVID-19, coupled with a $Z_\alpha$ of 1.96 for a two-tailed test, $Z_\beta$ at 0.84 for 80% power and a margin of error (E) of 5%, the sample size was estimated to be 106 for each study group giving rise to 212 participants for the two groups in each study site. This implies that for both CCTH and TTH, a total of 424 participants would have been adequate for the study. Ultimately, in the current analysis, data from 510 participants made up of 220 from Cape Coast and 290 from Tamale were used.

**Selection of study participants.** The simple random sampling technique was used to recruit 255 participants with past COVID-19 status and an equal number without past COVID-19 status, for the study. Fig 2 outlines the selection of study participants at the two study sites. Participants were recruited between 1st July, 2022 and 30th June, 2023 for the CCTH site. However, for the TTH arm, recruitment started from 1st March, 2023 to 30th November, 2023. Since management of COVID-19 in Ghana was reserved for selected teaching hospitals including the CCTH and TTH, records of such individuals could be assessed and contacted for the study. Therefore, 150 and 200 recovered COVID-19

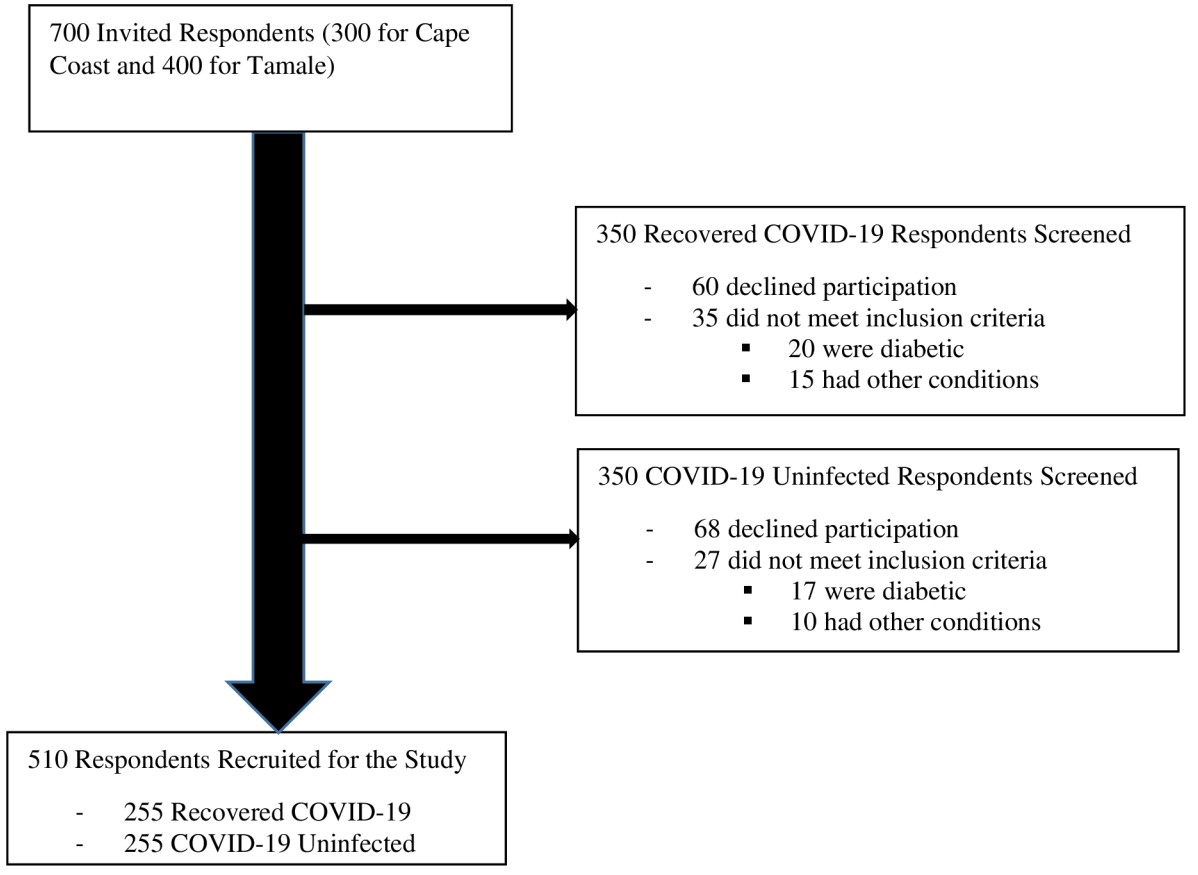

**Fig 2. Flow chart for participants' selection.**

participants were randomly selected from the CCTH and TTH respectively and contacted by telephone calls for participation in the study. Similarly, 150 and 200 control participants were selected from the Cape Coast and the Tamale metropolitan areas in a random manner in line with the study protocol for participation. After appropriate screening to ensure compliance with inclusion and exclusion criteria, 220 and 290 participants from the Cape Coast and Temale metropolitan areas respectively were recruited for the study. Details of the recruitment are shown below in Fig 2.

## Assessment of study variables

To achieve the objectives of the study, the following variables of demographic, anthropometric and biochemical relevance were assessed.

**Demographic variables.** Demographic information including age, education, past COVID-19 status, smoking and alcohol consumption status were collected by a structured questionnaire designed for the study.

**Blood pressure and anthropometric measurements.** An experienced nurse measured the blood pressure of the participant with an electronic sphygmomanometer (Contec Medical Systems Co. Ltd, China) in a routine manner. The blood pressure was measured on the right arm of participants in sitting position after a minimum of five minutes rest. An average of two measurements that did not differ by > 5 mmHg was recorded for each participant as the blood pressure.

Anthropometric indices such as weight (kg), height (m), waist circumference (cm) and hip circumference (cm) were measured for computation of waist-to-hip ratio and body mass index (BMI). Weight was measured to the nearest 0.1 kg with height to the nearest 0.1 cm. BMI was calculated by dividing the weight in kilogramme by the square of the height

in metre (kg/m²). Weight and height were measured in light clothing without footwear. Weight was measured by an electronic scale and the height by stadiometer (Seca mechanical column scale with stadiometer, USA). Waist and hip circumferences were measured in centimetres with an inflexible tape measure. Waist circumference was measured at the midpoint between the lower margin of the last rib and the top of the iliac crest but the hip circumference was measured around the widest portion of the buttocks. Waist-to-hip ratio was obtained by dividing the waist circumference by the hip circumference.

**Measurement of glucose, lipid profile, insulin and C-reactive protein levels.** Measurements of plasma glucose and lipid profile including LDL, HDL, total cholesterol, triglyceride and VLDL were done with the Mindray BS240 automated chemistry analyser (Mindray Diagnostics, Nanshan Shenzhen, China) following a routine procedure.

However, serum insulin and C-reactive protein (CRP) levels were determined by immunoturbidimetric test kits purchased from Kamiya Biomedicals Company (K-ASSAY Seattle, USA) in accordance with the manufacturer's detailed instructions outlined by Adatsi et al. [13].

**Assessment of insulin resistance and beta-cell function.** Insulin resistance was assessed by the homeostatic model, $HOMA-IR = Glucose\ \left(\frac{mmol}{l}\right) * Insulin\ (m\frac{U}{l})/22.5$ [14] and the triglyceride-glucose (TyG) index, $TyG = In\left[FBS\left(\frac{mg}{dl}\right) * TG\left(\frac{mg}{dl}\right)\right]/2$ [15]. Additionally, beta-cell function was determined using the homeostatic model, $HOMAB = 20 * Fasting\ Insulin\ \left(\frac{mU}{l}\right)/\ Fasting\ Glucose\ \left(\frac{mmol}{l}\right) - 3.5$ developed by Matthews et al. [14].

## Determination of abnormal levels of blood-based parameters

Dyslipidemias was assessed in accordance with the National Cholesterol Education Program Adult Treatment Panel (NCEP-ATP) III guidelines [16]. Specifically, abnormal levels of the lipid indices were defined as triglycerides ≥ 1.7 mmol/l; low-density lipoprotein cholesterol (LDL) ≥ 2.6 mmol/l; high-density lipoprotein cholesterol (HDL) < 1.03 mmol/l; total cholesterol ≥ 5.2 mmol/l. In addition, high levels of the other indices were defined as glucose > 6.4 mmol/l; insulin > 24.9 mIU/l; insulin resistance, HOMA-IR > 2.6 or TyG index > 4.0; beta-cell function, HOMA-B, high > 100% or low < 25%.

## Ethical approval

The Ethical Review Committee of the Tamale Teaching Hospital (TTHERC/24/02/23/01) granted approval for the study at Tamale as Cape Coast Teaching Hospital (CCTHERC/EC/2022/111) approved that for the Cape Coast arm. A written informed consent was provided by each study participant and at all times, the study protocols adhered strictly to the ethical standards of the Ghana Health Service and the World Medical Association Declaration of Helsinki.

## Data analysis

Statistical Software for Data Science (STATA) version 15 Corp (StataCorp LLC, USA) software was used to analyze the data for this study. Demographic characteristics and other categorical variables were presented in frequencies and percentages and compared with the chis-square goodness of fit test. Continuous variables were presented as mean and standard deviation if normally distributed or median and interquartile ranges if distribution deviated from normal distribution. The independent sample t-test was used to compare mean levels of measured indices between participants with and without past COVID-19 status for continuous variables that met assumptions for parametric test else the Mann Whitney test was used. Poisson regression with prevalence ratio was used to examine the relationship between insulin resistance, measured by HOMAIR or TyG, as the dependent variable and the other predictor variables. The univariate followed by the multivariate regression analyses were applied. Before fitting the multivariate regression model, the variance inflation factor (VIF) and tolerance (1/VIF) were used to assess multicollinearity among the independent variables, and no evidence of collinearity was found. The results of the regression analyses were presented as crude prevalence ratios (CPRs) and adjusted prevalence ratios (APRs) with 95% confidence intervals (CIs) and corresponding p-values. The determination of statistical significance for all analyses was considered at p < 0.05. Effect size

(Cohen's d) was computed for the adjusted models and the mean/median comparisons to examine the strengths of the relationships. The standard interpretations for effect size were adopted in interpreting our data.

## Results

In the current study, two different surrogate markers (HOMA-IR and TyG) were used for the assessment of insulin resistance to allow for probable identification of variety of promoters of the condition in our context. Various demographic, anthropometric and blood-based indices known to be associated with insulin resistance in various populations were assessed in the current study that involved 510 residents of Tamale and Cape Coast metropolis.

Data from 290 participants from Tamale (150 females and 140 males) and 220 participants from Cape Coast (123 males and 97 females) were used in our analysis.

### Sociodemographic factors

The sociodemographic characteristics of participants (Table 1) indicate that a significant (P < 0.05; Table 1) majority of the participants were residents of Tamale (56.86%), aged 20–39 years (69.02%), had tertiary education (65.29%),

**Table 1. Sociodemographic characteristics of participants.**

| Parameter | Frequency (%) | P-value |
|---|---|---|
| **Location** | | 0.002* |
| Cape Coast | 220 (43.14) | |
| Tamale | 290 (56.86) | |
| **Gender** | | 0.48 |
| Female | 247 (48.43) | |
| Male | 263 (51.57) | |
| **Age** | | < 0.001* |
| 20-29 | 189 (37.06) | |
| 30-39 | 163 (31.96) | |
| 40-49 | 92 (18.04) | |
| 50-59 | 36 (7.06) | |
| 60 and above | 30 (5.88) | |
| **Education** | | < 0.001* |
| No education | 75 (14.71) | |
| Primary | 24 (4.71) | |
| Secondary | 81 (15.88) | |
| Tertiary | 330 (64.71) | |
| **Past COVID-19** | | 1.00 |
| No | 255 (50) | |
| Yes | 255 (50) | |
| **Smoking status** | | < 0.001* |
| No | 489 (95.88) | |
| Yes | 21 (4.12) | |
| **Alcohol consumption** | | < 0.001* |
| No | 388 (76.08) | |
| Yes | 122 (23.92) | |

**COVID-19:** Coronavirus disease 2019; **\*:** Significant p-value for chi-square goodness of fit test

**Table 2. Levels of measured parameters by COVID-19 status.**

| Parameter | Entire participants Mean±SD | COVID-19 uninfected Mean±SD | Recovered COVID-19 Mean±SD | P-value | Effect size (Cohen's d) |
|---|---|---|---|---|---|
| Weight (kg) | 71.73±14.28 | 70.64±14.60 | 72.83±13.89 | 0.08 | 0.15 |
| Height (m) | 1.68±0.09 | 1.68±0.10 | 1.68±0.08 | 0.87 | 0.01 |
| Waist circumference (cm) | 82.61±11.67 | 81.46±11.92 | 83.75±11.33 | 0.03* | 0.20 |
| Hip circumference (cm) | 95.80±13.06 | 94.31±0.85 | 97.28±0.79 | 0.01* | 0.23 |
| WHR | 0.86±0.06 | 0.87±0.06 | 0.86±0.06 | 0.52 | 0.06 |
| Systolic BP (mmHg) | 126.33±14.83 | 126.24±15.47 | 126.41±14.19 | 0.9 | 0.01 |
| BMI (kg/m$^2$) | 25.43±5.07 | 25.12±4.98 | 25.75±5.14 | 0.16 | 0.15 |
| FPG (mmol/L) | 4.74±0.95 | 4.53±0.81 | 4.94±1.04 | <0.001* | 0.44 |
| | Median (IQR) | Median (IQR) | Median (IQR) | | |
| Insulin (mIU/L) | 18.45 (13.4-27.5) | 17.80 (12.40-23.40) | 19.4 (14.1-45.6) | <0.001* | 0.17 |
| HOMA-B (%) | 115.93 (68.9-295.60) | 111.71 (66.09-253.04) | 127.19 (69.57-372.50) | 0.12 | 0.07 |
| HOMAIR (mIU/L) | 3.72 (2.57-5.99) | 3.43 (2.47-4.74) | 4.25 (2.78-10.63) | <0.001* | 0.20 |
| TyG index | 4.46 (4.26-4.71) | 4.45 (4.25-4.66) | 4.49 (4.33-4.73) | 0.26 | 0.05 |
| HDL (mmol/L) | 1.73 (1.32-2.45) | 1.70 (1.32-2.33) | 1.74 (1.29-2.59) | 0.39 | 0.04 |
| LDL (mmol/L) | 2.46 (1.62-3.48) | 2.30 (1.48-3.38) | 2.81 (1.78-3.54) | 0.03* | 0.10 |
| VLDL(mmol/L) | 0.43 (0.33-0.7) | 0.39 (0.33-0.66) | 0.46 (3.13-72.4) | 0.9 | 0.07 |
| Total cholesterol (mmol/L) | 4.42 (3.24-5.32) | 4.21 (3.02-5.32) | 4.56 (3.67-5.32) | 0.04* | 0.10 |
| Triglyceride (mmol/L) | 0.99 (0.71-1.62) | 1.01 (0.72-1.52) | 0.97 (0.71-1.69) | 0.74 | 0.02 |
| CRP (mg/dL) | 1.00 (0.4-1.4) | 0.80 (0.30-1.20) | 1.20 (0.70-1.78) | <0.001* | 0.26 |

**CRP:** C-reactive protein; **BMI:** Body mass index; **FPG:** Fasting plasma glucose; **HDL:** High-density lipoprotein cholesterol; **VLDL:** Very low-density lipoprotein cholesterol; **HOMA-IR:** Homeostatic model assessment of insulin resistance; **HOMAB:** Homeostatic model of beta-cell function; **BP:** Blood pressure; **WHR:** Weight-to-height ratio; **SD:** Standard deviation; **IQR:** Inter-quartile range; **COVID-19:** Coronavirus disease 2019; **\*:** Significant p-value for independent sample t-test (mean) or Mann Whitney test (median).

non-smokers (95.88%) and did not drink alcohol (76.08%). In terms of gender, proportions of male and female were similar statistically (P = 0.48; Table 1).

## Mean/median levels of measured indices

Compared to the uninfected group, the recovered COVID-19 participants had higher (P < 0.05) levels of waist circumference, hip circumference, fasting plasma glucose, insulin, low-density lipoprotein cholesterol, total cholesterol, HOMA-IR and C-reactive protein (Table 2) although the strengths of the observed associations were generally small.

## Abnormal levels of measured variables

Varied proportions (Fig 3) of abnormal levels of the measured indices were observed among the study participants. For example, the prevalence of insulin resistance (IR) ranged from 70.69% (Tamale) to 79.09% (Cape Coast) for the HOMA-IR index with an overall rate of 74.31% (Table 3 and Fig 3). With TyG index, the observed IR prevalence were 88.62% and 90.91% (Table 4) respectively for the Tamale and Cape Coast residents with an overall rate of 89.61% (Fig 3). However, the respective IR prevalence of 70.98% and 88% (HOMA-IR; Table 3) and 89.02% and 90.2% (TyG; Table 4) for participants without past COVID-19 status and those with past COVID-19 status were observed. Irrespective of location or COVID-19 status, low (31.18%) and high (19.41%) levels of beta-cell function were also observed (Fig 3). In terms of dyslipidemias, prevalence values of 8.43% for very low-density lipoprotein cholesterol, high levels of very-low density lipoprotein cholesterol (8.31%), triglycerides (24.9%), total cholesterol (27.45%), low-density lipoprotein cholesterol

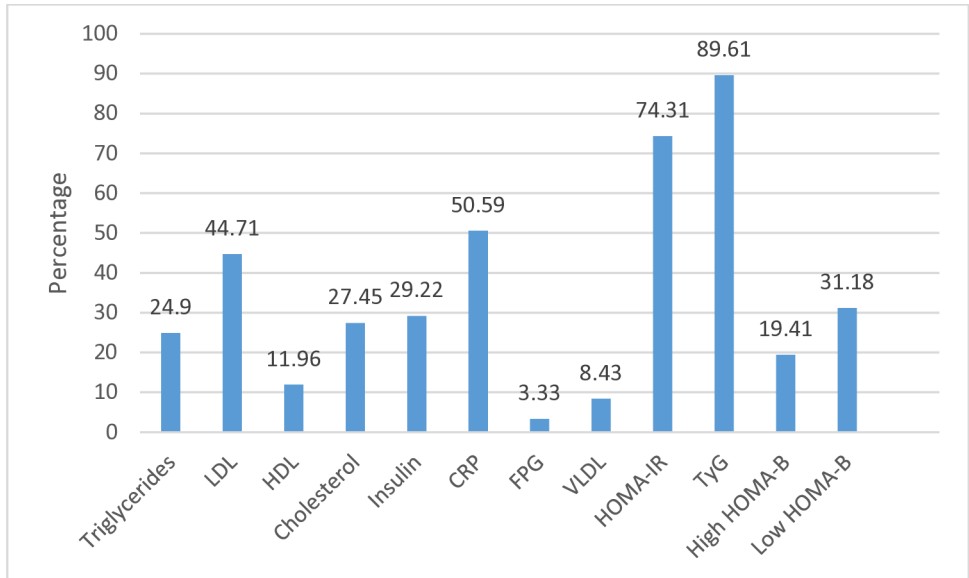

**Fig 3. Prevalence of abnormal levels of blood-based indices.** *Except otherwise indicated, all the proportions represent percentage of individuals with levels of measured indices above the upper threshold of the reference range apart from HDL which represents a level below the lower threshold of its reference range.* **CRP:** *C-reactive protein;* **BMI:** *Body mass index;* **FPG:** *Fasting plasma glucose;* **HDL:** *High-density lipoprotein cholesterol;* **VLDL:** *Very low-density lipoprotein cholesterol;* **HOMA-IR:** *Homeostatic model assessment of insulin resistance;* **HOMAB:** *Homeostatic model of beta-cell function;* **LDL:** *low-density lipoprotein cholesterol;* **TyG:** *triglyceride-glucose index.*

(44.71%) and low level of high-density lipoprotein cholesterol (11.96%) and low-grade inflammation (50.59%) were noted (Fig 3).

### Predictors of insulin resistance determined by HOMA-IR

In a regression analysis to identify predictors of insulin resistance using HOMA-IR as the dependent variable in the entire sample irrespective of past COVID-19 status, age, education and triglycerides came out as the only predictors of insulin resistance (Table 3). Residing in Tamale decreased the prevalence of insulin resistance by 11% (PR: 0.89; CI: 0.81-0.99; P = 0.03; Table 3) compared to being a resident of Cape Coast in a univariate analysis. However, this reduction could not be sustained after adjusting for confounders (PR: 0.96; CI: 0.91-1.02; P = 0.23; Table 3) in a multivariate analysis although 79.09% (174/220) of Cape Coast residents and 70.69% (205/290) of Tamale residents were insulin resistant. Compared with age 20–29 years, participants in age group 30–39 and 50–59 years had respective increased prevalence of 18% (PR: 1.18; CI: 1.03-1.33; P = 0.01; Table 3) and 27% (PR: 1.27; CI: 1.07-1.50; P = 0.004; Table 3) in having insulin resistance with no such effect observed for those in age group 40–49 or 60 years and above. After controlling for confounders, only the participants in age group 50–59 years demonstrated 11% increase in prevalence of insulin resistance (PR: 1.11; CI: 1.02-1.21; P = 0.01; Table 3). The strength of the observed associations ranged from small (30–39 years group) to medium (50–59 years group). Having high glucose level did not demonstrate statistically significant association with insulin resistance although high triglycerides increased the prevalence by 5% (PR: 1.05; CI: 1.00-1.11; P = 0.04; Table 3) with medium strength of the observed association. Interestingly, the prevalence of insulin resistance increased by 13% (PR: 1.13; CI: 1.01-1.27; P = 0.03; Table 3) in participants with secondary education but decreased by 98% (PR: 0.02; CI: 0.002-0.14; P < 0.001; Table 3) in those with tertiary education in our setting. Indeed, across levels, the strength of the association between insulin resistance and formal education in general, was large in our context. Statistically, neither participants with high insulin levels (PR: 1.18; CI: 0.75-1.85; P = 0.46; Table 3) nor past COVID-19 status (PR: 0.98; CI: 0.92-1.04; P = 0.56;

**Table 3. Predictors of insulin resistance determined by HOMA-IR.**

| Parameter | No insulin resistance, N (%) | Insulin resistance, N (%) | CPR (95% CI) | P-value | APR (95%CI) | P-value | Effect size (Cohen's d) |
|---|---|---|---|---|---|---|---|
| **Location** | | | | | | | |
| Cape Coast (ref.) | 46 (35.11) | 174 (45.91) | 1 | | 1 | | |
| Tamale | 85 (64.89) | 205 (54.09) | 0.89(0.81-0.99) | 0.03* | 0.96(0.91-1.02) | 0.23 | -0.24 |
| **Gender** | | | | | | | |
| Female (ref.) | 57 (43.51) | 190 (50.13) | 1 | | 1 | | |
| Male | 74 (56.49) | 189 (49.87) | 0.93(0.84-1.03) | 0.19 | 1.04(0.99-1.10) | 0.11 | -0.18 |
| **Age** | | | | | | | |
| 20-29 (ref) | 61 (46.57) | 128 (33.77) | 1 | | 1 | | |
| 30-39 | 33 (25.19) | 130 (34.30) | 1.18(1.03-1.33) | 0.01* | 1.03(0.96-1.10) | 0.35 | 0.29 |
| 40-49 | 22 (16.79) | 70 (18.47) | 1.12(0.97-1.31) | 0.15 | 1.06(0.98-1.15) | 0.2 | 0.21 |
| 50-59 | 5 (3.82) | 31 (8.18) | 1.27(1.07-1.50) | 0.004* | 1.11(1.02-1.21) | 0.01* | 0.64 |
| 60 and above | 10 (7.63) | 20 (5.28) | 0.98(0.75-1.29) | 0.91 | 0.97(0.84-1.12) | 0.71 | -0.09 |
| **Education** | | | | | | | |
| No education (ref.) | 1 (0.76) | 75 (19.79) | 1 | | 1 | | |
| Primary | 3 (2.29) | 20 (5.28) | 0.73(0.51-1.06) | 0.11 | 1.03(0.85-1.26) | 0.73 | -1.08 |
| Secondary | 9 (6.87) | 72 (18.99) | 0.83(0.69-1.00) | 0.06 | 1.13(1.01-1.27) | 0.03* | -1.17 |
| Tertiary | 118 (90.08) | 212 (55.94) | 0.95(0.83-1.08) | 0.44 | 0.02(0.002-0.14) | <0.001* | -2.16 |
| **Smoke** | | | | | | | |
| No (ref.) | 126 (96.18) | 363 (95.78) | 1 | | 1 | | |
| Yes | 5 (3.82) | 16 (4.22) | 1.02(0.80-1.31) | 0.84 | 0.96(0.83-1.11) | 0.61 | 0.07 |
| **Past COVID-19** | | | | | | | |
| No (ref.) | 74 (56.49) | 181 (47.76) | 1 | | 1 | | |
| Yes | 57 (43.51) | 198 (52.24) | 1.09(0.98-1.21) | 0.09 | 0.98(0.92-1.04) | 0.56 | 0.23 |
| **HOMA-B** | | | | | | | |
| Normal (ref.) | 7 (5.34) | 245 (64.64) | 1 | | 1 | | |
| High | 38 (29.01) | 61 (16.09) | 1.45(0.76-1.56) | 0.39 | 1.08(0.18-1.29) | 0.15 | -0.39 |
| Low | 86 (65.65) | 73 (19.26) | 0.99(0.89-1.20) | 0.67 | 0.98(0.70-1.14) | 0.23 | -1.65 |
| **Waist-to-hip ratio** | | | | | | | |
| Normal (ref.) | 71 (54.20) | 219 (57.78) | 1 | | 1 | | |
| High | 60 (45.80) | 160 (42.22) | 0.91 (0.61-1.36) | 0.63 | 0.86(0.57-1.31) | 0.49 | -0.08 |
| **Waist circumference** | | | | | | | |
| Normal (ref.) | 108 (82.44) | 321 (84.70) | 1 | | 1 | | |
| High | 23 (17.56) | 58 (15.30) | 0.96(0.82-1.11) | 0.59 | 0.93(0.85-1.02) | 0.14 | -0.13 |
| **BMI** | | | | | | | |
| Normal (ref.) | 100 (76.34) | 210 (55.41) | 1 | | 1 | | |
| Underweight | 10 (7.63) | 15 (3.96) | 0.91(0.64-1.30) | 0.62 | 1.01(0.79-1.31) | 0.87 | 0.18 |
| Overweight | 16 (12.21) | 115 (30.34) | 1.02(0.71-1.46) | 0.90 | 1.09(0.85-1.40) | 0.47 | 0.22 |
| Obese | 5 (3.82) | 39 (10.29) | 0.94(0.63-1.39) | 0.74 | 1.16(0.90-1.51) | 0.23 | 0.06 |
| **FPG** | | | | | | | |
| Normal (ref.) | 129 (92.37) | 363 (94.72) | 1 | | 1 | | |
| High | 2 (1.53) | 16 (4.22) | 2.67 (1.18-6.03) | 0.02* | 1.32(0.90-1.94) | 0.15 | 0.68 |
| **Insulin** | | | | | | | |
| Normal (ref.) | 130 (99.24) | 231 (60.95) | 1 | | 1 | | |
| High | 1 (0.76) | 148 (39.05) | 1.23(0.78-1.89 | 0.45 | 1.18(0.75-1.85) | 0.46 | 2.43 |

*(Continued)*

**Table 3.** (Continued)

| Parameter | No insulin resistance, N (%) | Insulin resistance, N (%) | CPR (95% CI) | P-value | APR (95%CI) | P-value | Effect size (Cohen's d) |
|---|---|---|---|---|---|---|---|
| **LDL** | | | | | | | |
| Normal (ref) | 82 (62.60) | 200 (52.77) | 1 | | 1 | | |
| High | 50 (37.40) | 179 (47.23) | 1.04 (0.93-1.17) | 0.48 | 0.96(0.89-1.02) | 0.24 | 0.07 |
| **HDL** | | | | | | | |
| Normal | 114 (87.02) | 336 (88.65) | 1 | | 1 | | |
| Low | 17 (12.98) | 43 (11.35) | 1.15 (0.63-2.09) | 0.65 | 1.14(0.61-2.13) | 0.67 | 0.07 |
| **VLDL** | | | | | | | |
| Normal (ref.) | 120 (91.60) | 347 (91.56) | 1 | | 1 | | |
| High | 11 (8.40) | 32 (8.44) | 1.00(0.83-1.20) | 0.96 | 0.97(0.91-1.04) | 0.43 | -0.02 |
| **Cholesterol** | | | | | | | |
| Normal (ref.) | 96 (73.28) | 274 (72.30) | 1 | | 1 | | |
| High | 35 (26.72) | 105 (27.70) | 1.01 (0.91-1.14) | 0.77 | 1.04(0.98-1.10) | 0.18 | 0.02 |
| **Triglyceride** | | | | | | | |
| Normal (ref.) | 103 (78.63) | 17 (4.49) | 1 | | 1 | | |
| High | 28 (21.37) | 362 (95.51) | 1.28(0.79-2.06) | 0.32 | 1.05(1.00-1.11) | 0.04* | 0.51 |
| **CRP** | | | | | | | |
| Normal (ref.) | 70 (53.44) | 182 (48.02) | 1 | | 1 | | |
| High | 61 (46.56) | 197 (51.19) | 1.05(0.95-1.17) | 0.32 | 0.95(0.88-1.03) | 0.24 | 0.13 |

**CRP:** C-reactive protein; **BMI:** Body mass index; **FPG:** Fasting plasma glucose; **HDL:** High-density lipoprotein cholesterol; **VLDL:** Very low-density lipoprotein cholesterol; **HOMA-IR:** Homeostatic model assessment of insulin resistance; **HOMA-B:** Homeostatic model of beta-cell function; **WHR:** Weight-to-height ratio; **N:** Number of participants; **COVID-19:** Coronavirus disease 2019; **CPR:** Crude prevalence ratio (univariate); **APR:** Adjusted prevalence ratio (multivariate); *: Significant p-value

Table 3) demonstrated increased prevalence of insulin resistance compared to their counterparts with normal levels of the parameter or without past COVID-19 status in our context.

## Predictors of insulin resistance determined by TyG

In a similar analysis with TyG (triglyceride-glucose index) as the measure of insulin resistance, age, education and tri-glycerides (P<0.05; Table 4) were the only parameters that could predict insulin resistance with the strength of the association ranging from small (education) to medium (age and triglycerides). Participants in age group 50–59 years showed a 12% increased prevalence of insulin resistance (PR: 1.12; CI: 1.02-1.21; P=0.01; Table 4) with medium strength of the observed association. Similarly, those with tertiary level educational attainment had a 12% increased prevalence (PR: 1.12; CI: 1.01-1.26; P=0.04; Table 4) of insulin resistance but the strength of the observed association was small. The prevalence of insulin resistance increased by only 6% (PR: 1.06; CI: 1.00-1.11; P=0.04; Table 4) for participants with high triglycerides levels although the association demonstrated a medium strength. Interestingly, as shown in Table 4, high levels of beta-cell function (PR: 0.97; CI: 0.942-1.76; P=0.8), FPG (PR: 1.32; CI: 0.91-1.94; P=0.15), CRP (PR: 0.96; CI: 0.88-1.03; P=0.24); insulin (PR: 1.16; CI: 0.74-1.83; P=0.52), and past COVID-19 status (PR: 0.98; CI: 0.92-1.05; 0.56) did not reveal any statistically significant association with insulin resistance.

## Predictors of insulin resistance measured by HOMA-IR according to past COVID-19 status

In a sub-group analysis based on past COVID-19 status, only the levels of fasting plasma glucose, insulin and total cholesterol could significantly (P<0.05; Table 5) associate with insulin resistance in participants without past COVID-19 status.

**Table 4. Predictors of insulin resistance determined by triglyceride-glucose (TyG) index.**

| Parameter | No insulin resistance, N (%) | Insulin resistance, N (%) | CPR (95%CI) | P-value | APR (95%CI) | P-value | Effect size (Cohen's d) |
|---|---|---|---|---|---|---|---|
| **Location** | | | | | | | |
| Cape Coast (ref.) | 20 (37.74) | 200 (43.76) | 1 | | 1 | | |
| Tamale | 33 (62.26) | 257 (56.24) | 0.95(0.90-1.01) | 0.08 | 0.96(0.91-1.02) | 0.23 | -0.47 |
| **Sex** | | | | | | | |
| Female (ref.) | 30 (56.60) | 217 (47.48) | 1 | | 1 | | |
| Male | 23 (43.40) | 240 (52.52) | 1.02(0.97-1.09) | 0.30 | 1.05(0.91-1.10) | 0.11 | 0.27 |
| **Age** | | | | | | | |
| 20-29 (ref) | 22 (41.51) | 167 (36.54) | 1 | | 1 | | |
| 30-39 | 12 (22.64) | 151 (33.04) | 1.06(0.99-1.13) | 0.10 | 1.03(0.96-1.11) | 0.96 | 0.19 |
| 40-49 | 7 (13.21) | 85 (18.60) | 1.07(0.99-1.16) | 0.06 | 1.07(0.98-1.16) | 0.18 | 0.25 |
| 50-59 | 8 (15.09) | 28 (6.13) | 1.11(1.02-1.19) | 0.01* | 1.12(1.02-1.21) | 0.01* | 0.50 |
| 60 and above | 4 (7.55) | 26 (5.69) | 0.98(0.84-1.15) | 0.85 | 0.97(0.84-1.13) | 0.71 | -0.33 |
| **Education** | | | | | | | |
| No education (ref.) | 7 (13.21) | 69 (15.10) | 1 | | 1 | | |
| Primary | 7 (13.21) | 16 (3.50) | 1.02(0.84-1.23) | 0.87 | 1.03(0.85-1.26) | 0.73 | 0.03 |
| Secondary | 12 (22.64) | 69 (15.10) | 1.12(1.00-1.25) | 0.0.04* | 1.13(1.01-1.27) | 0.23 | 0.45 |
| Tertiary | 27 (50.94) | 303 (66.30) | 1.09(0.98-3.02) | 0.11 | 1.12(1.01-1.26) | 0.04* | 0.07 |
| **Smoke** | | | | | | | |
| No (ref.) | 51 (96.23) | 438 (95.84) | 1 | | 1 | | |
| Yes | 2 (3.77) | 19 (4.16) | 0.99(0.89-1.14) | 0.77 | 0.96(0.83-1.11) | 0.61 | -0.06 |
| **Past Covid-19** | | | | | | | |
| No (ref.) | 25 (47.17) | 230 (50.33) | 1 | | 1 | | |
| Yes | 28 (52.83) | 227 (49.67) | 1.00(0.95-1.06) | 0.87 | 0.98(0.92-1.05) | 0.56 | -0.33 |
| **HOMA-B** | | | | | | | |
| Normal (ref.) | 31 (58.49) | 221 (48.36) | 1 | | 1 | | |
| High | 10 (18.87) | 89 (19.47) | 1.11(0.99-1.24) | 0.06 | 0.97(0.942-1.76) | 0.8 | -0.49 |
| Low | 12 (22.64) | 147 (32.17) | 1.09(0.97-1.22) | 0.14 | 0.96(0.94-1.21) | 0.32 | -0.11 |
| **Waist-to-hip ratio** | | | | | | | |
| Normal (ref.) | 35 (66.04) | 261 (57.11) | 1 | | 1 | | |
| High | 18 (33.96) | 196 (42.89) | 1.12(0.98-1.09) | 0.20 | 0.92(0.91-1.21) | 0.61 | 0.01 |
| **Waist circumference** | | | | | | | |
| Normal (ref.) | 48 (90.57) | 381 (83.37) | 1 | | 1 | | |
| High | 5 (9.43) | 76 (16.63) | 1.03(0.96-1.10) | 0.38 | 0.93(0.85-1.02) | 0.14 | -0.68 |
| **BMI** | | | | | | | |
| Normal (ref.) | 15 (2.56) | 295 (64.55) | 1 | | 1 | | |
| Underweight | 14 (74.36) | 11 (2.41) | 0.99(0.78-1.25) | 0.93 | 1.02(0.80-1.31) | 0.88 | 0.2 |
| Overweight | 19 (17.95) | 109 (23.85) | 1.07(0.85-1.36) | 0.55 | 1.09(0.85-1.40) | 0.48 | -0.01 |
| Obese | 5 (5.13) | 42 (9.19) | 1.07(0.84-1.36) | 0.57 | 1.17(0.90-1.51) | 0.24 | 0.56 |
| **FPG** | | | | | | | |
| Normal (ref.) | 53 (92.45) | 439 (96.06) | 1 | | 1 | | |
| High | 0 (0.00) | 18 (3.94) | 1.42(0.93-2.15) | 0.10 | 1.32(0.91-1.94) | 0.15 | 0.47 |
| **Insulin** | | | | | | | |
| Normal (ref.) | 52 (98.11) | 309 (67.62) | 1 | | 1 | | |

*(Continued)*

**Table 4.** (Continued)

| Parameter | No insulin resistance, N (%) | Insulin resistance, N (%) | CPR (95%CI) | P-value | APR (95%CI) | P-value | Effect size (Cohen's d) |
|---|---|---|---|---|---|---|---|
| High | 1 (1.89) | 148 (32.38) | 1.21(0.68-2.13) | 0.65 | 1.16(0.74-1.83) | 0.52 | 1.21 |
| **LDL** | | | | | | | |
| Normal (ref) | 39 (73.59) | 243 (72.87) | 1 | | 1 | | |
| High | 14 (26.41) | 214 (27.13) | 1.03(0.97-1.09) | 0.36 | 0.96(0.89-1.03) | 0.25 | 0.22 |
| **HDL** | | | | | | | |
| Normal | 47 (88.68) | 402 (87.96) | 1 | | 1 | | |
| Low | 6 (11.32) | 55 (12.04) | 0.74 (0.30-1.14) | 0.48 | 0.45 (0.21-1.10) | 0.28 | -0.32 |
| **VLDL** | | | | | | | |
| Normal (ref.) | 42 (71.79) | 425 (93.00) | 1 | | 1 | | |
| High | 11 (28.21) | 32 (7.00) | 1.04(0.97-1.13) | 0.22 | 1.04(0.91-1.04) | 0.43 | 0.02 |
| **Total cholesterol** | | | | | | | |
| Normal (ref.) | 18 (10.26) | 352 (70.46) | 1 | | 1 | | |
| High | 35 (89.74) | 105 (29.54) | 1.07(1.03-1.13) | 0.002 | 1.04(0.98-1.11) | 0.18 | 0.68 |
| **Triglyceride** | | | | | | | |
| Normal (ref.) | 39 (73.59) | 344 (75.27) | 1 | | 1 | | |
| High | 14 (26.41) | 113 (24.73) | 1.07(1.02-1.12) | 0.03* | 1.06(1.00-1.11) | 0.04* | 0.63 |
| **CRP** | | | | | | | |
| Normal (ref.) | 19 (35.85) | 233 (50.98) | 1 | | 1 | | |
| High | 34 (64.15) | 224 (49.02) | 1.07(1.03-1.010) | <0.001 | 0.96(0.88-1.03) | 0.24 | -0.49 |

**CRP:** C-reactive protein; **BMI:** Body mass index; **FPG:** Fasting plasma glucose; **HDL:** High-density lipoprotein cholesterol; **VLDL:** Very low-density lipoprotein cholesterol; **HOMA-IR:** Homeostatic model assessment of insulin resistance; **HOMA-B:** Homeostatic model of beta-cell function; **WHR:** Weight-to-height ratio; **N:** Number of participants; **COVID-19:** Coronavirus disease 2019; **CPR:** Crude prevalence ratio (univariate); **APR:** Adjusted prevalence ratio (multivariate); **\*:** Significant p-value

Participants with high levels of plasma glucose (PR: 2.06; 1.03-4.10; P = 0.03; Table 5) and insulin (PR: 1.57; CI: 1.38-1.79; P < 0.001; Table 5) exhibited 106% and 57% respective increased prevalence of insulin resistance compared with their counterparts with normal levels of the parameters. The strengths of the observed associations ranged from medium (insulin) to high (FPG). However, participants with high total cholesterol level showed a 24% reduction in prevalence of insulin resistance (PR: 0.76; CI: 0.61-0.95; P = 0.01; Table 5) with a small strength of the observed association. The seeming 19% reduction in prevalence of insulin resistance by residing in Tamale (PR: 0.81; CI: 0.70-0.95; P = 0.01) observed in the univariate analysis in participants without past COVID-19 status could not be sustained after controlling for confounders in the multivariate analysis (PR: 0.87; CI: 0.72-1.07; P = 0.19; Table 5). Compared to age 20–29 years, age 30–39 years had a 124% increased prevalence of insulin resistance (PR: 2.24, CI: 1.12-4.49; P = 0.02; Table 5) in participants without past COVID-19 status but this effect vanished after adjustment (APR: 1.13; CI: 0.94-1.30; P = 0.25; Table 5). Interestingly, none of the measured indices exhibited a statistically significant (P > 0.05; Table 5) association with insulin resistance in the participants with past COVID-19 status.

## Predictors of insulin resistance measured by TyG according to past COVID-19 status

In a sub-group analysis with TyG as the index for insulin resistance based on past COVID-19 status, age, BMI, triglycerides and CRP turned out as the statistically significant (P < 0.05; Table 6) predictors of insulin resistance in participants with past COVID-19 status. For participants without past COVID-19 status, age emerged as the sole predictor of

**Table 5. Predictors of insulin resistance measured by HOMA-IR according to past COVID-19 status.**

| Parameter | PAST COVID-19 STATUS | | | | | NO PAST COVID-19 STATUS | | | | |
|---|---|---|---|---|---|---|---|---|---|---|
| | CPR (95%CI) | P-value | APR (95%CI) | P-value | ES | CPR (95%CI) | P-value | APR (95%CI) | P-value | ES |
| **Location** | | | | | | | | | | |
| Cape Coast **(ref.)** | 1 | | 1 | | | 1 | | 1 | | |
| Tamale | 0.96(0.84-1.10) | 0.63 | 0.99(0.87-1.13) | 0.95 | 0.20 | 0.81(0.70-0.95) | 0.01* | 0.87(0.72-1.07) | 0.19 | 0.35 |
| **Sex** | | | | | | | | | | |
| Females (ref.) | 1 | | 1 | | | 1 | | 1 | | |
| Males | 0.92(0.81-1.05) | 0.23 | 0.97(0.85-1.11) | 0.69 | 0.34 | 0.93(0.80-1.09) | 0.41 | 0.89(0.76-1.04) | 0.14 | 0.17 |
| **Age** | | | | | | | | | | |
| 20-29 **(ref)** | 1 | | 1 | | | 1 | | 1 | | |
| 30-39 | 1.50(0.75-3.02) | 0.25 | 1.10(0.93-1.30) | 0.38 | 0.10 | 2.24(1.12-4.49) | 0.02* | 1.13(0.94-1.30) | 0.25 | 0.39 |
| 40-49 | 1.10(0.93-1.33) | 0.30 | 1.08(0.90-1.30) | 0.36 | 0.24 | 1.42(0.65-3.11) | 0.38 | 1.10(0.91-1.33) | 0.34 | 0.17 |
| 50-59 | 1.18(0.92-1.52) | 0.18 | 1.11(0.84-1.47) | 0.84 | 0.10 | 3.61(0.99-13.00) | 0.05 | 1.18(0.92-1.52) | 0.18 | 0.78 |
| 60 and above | 1.10(0.83-1.46) | 0.49 | 1.02(0.77-1.35) | 0.88 | -0.07 | 0.65(0.22-1.94) | 0.44 | 1.10(0.83-1.46) | 0.49 | 0.49 |
| **Education** | | | | | | | | | | |
| No education (ref.) | 1 | | 1 | | | 1 | | 1 | | |
| Primary | 0.91(0.34-3.25) | 0.32 | 1.01(0.63-4.73) | 0.45 | 0.01 | 9.75(0.78-120.95) | 0.08 | 0.73(0.51-1.06) | 0.10 | 1.29 |
| Secondary | 0.30(0.11-1.82) | 0.51 | 0.46(0.21-2.34) | 0.32 | -0.43 | 4.10(0.44-38.42) | 0.22 | 0.83(0.68-1.00) | 0.06 | 0.79 |
| Tertiary | 0.28(0.17-2.13) | 0.21 | 0.33(0.18-3.00) | 0.12 | | 25.07(3.36-187.12 | 0.002* | 0.87(0.72-1.05) | 0.16 | 1.82 |
| **Smoke** | | | | | | | | | | |
| No **(ref.)** | 1 | | 1 | | | 1 | | 1 | | |
| Yes | 0.87(0.62-1.23) | 0.38 | 0.88(0.64-2.22) | 0.46 | -0.29 | 0.89(0.20-1.84) | 0.38 | 0.78(0.25-1.90) | 0.41 | -0.14 |
| **HOMA-B** | | | | | | | | | | |
| Normal (ref.) | 1 | | 1 | | | 1 | | 1 | | |
| High | 0.74(0.56-1.06) | 0.34 | 0.78(0.05-1.56) | 0.10 | -0.14 | 0.91(0.32-2.62) | 0.87 | 1.20(0.45-1.56) | 0.46 | 0.11 |
| Low | 0.56(0.4-1.24) | 0.64 | 0.06(0.03-1.10) | 0.13 | 0.11 | 0.12(0.04-1.35) | 0.56 | 0.56(0.06-1.11) | 0.34 | 0.01 |
| **WHR** | | | | | | | | | | |
| Normal **(ref.)** | 1 | | 1 | | | 1 | | 1 | | |
| High | 1.54(0.48-2.92) | 0.46 | 0.98(0.49-1.97) | 0.96 | -0.01 | 0.75(0.51-1.08) | 0.27 | 0.90(0.38-2.56) | 0.98 | -0.01 |
| **Waist circumference** | | | | | | | | | | |
| Normal (ref.) | 1 | | 1 | | | 1 | | 1 | | |
| High | 1.08(0.93-1.26) | 0.28 | 0.94(0.81-1.09) | 0.43 | 0.36 | 0.76(0.56-1.04) | 0.04* | 0.76(0.56-1.04) | 0.09 | -0.25 |
| **BMI** | | | | | | | | | | |
| Normal **(ref.)** | 1 | | 1 | | | 1 | | 1 | | |
| Underweight | 0.94(0.53-1.68) | 0.84 | 0.92(0.54-1.57) | 0.77 | 0.38 | 0.89(0.56-1.39) | 0.61 | 0.91(0.58-1.44) | 0.71 | -0.23 |
| Overweight | 1.15(0.65-2.04) | 0.62 | 1.05(0.62-1.79) | 0.62 | 0.25 | 0.89(0.56-1.41) | 0.62 | 0.90(0.56-1.43) | 0.66 | -0.34 |
| Obese | 1.06(0.57-4.71) | 0.83 | 0.99(0.56-1.75) | 0.98 | 0.19 | 0.78(0.45-1.37) | 0.40 | 0.99(0.53-1.84) | 0.98 | -0.58 |
| **FPG** | | | | | | | | | | |
| Normal (ref) | 1 | | 1 | | | 1 | | 1 | | |
| High | 0.91(0.73-1.14) | 0.44 | 1.50(0.42-4.87) | 0.34 | 0.22 | 1.42(0.71-1.87) | 0.32 | 2.06(1.03-4.10) | 0.03* | 0.86 |
| **Insulin** | | | | | | | | | | |
| Normal (ref.) | **1** | | 1 | | | 1 | | 1 | | |
| High | 0.94(0.74-1.22) | 0.34 | 1.92(0.98-3.98) | 0.09 | 0.06 | 1.54(1.37-1.72) | <0.001* | 1.57(1.38-1.79) | <0.001* | 0.67 |
| **LDL** | | | | | | | | | | |
| Normal (ref.) | 1 | | 1 | | | 1 | | 1 | | |
| High | 0.94(0.80-1.10) | 0.48 | 1.08(0.87-2.32) | 0.17 | -0.02 | 1.15(0.98-1.35) | 0.08 | 1.23(0.98-1.55) | 0.06 | 0.14 |

*(Continued)*

**Table 5.** (Continued)

| Parameter | PAST COVID-19 STATUS | | | | | NO PAST COVID-19 STATUS | | | | |
|---|---|---|---|---|---|---|---|---|---|---|
| | CPR (95%CI) | P-value | APR (95%CI) | P-value | ES | CPR (95%CI) | P-value | APR (95%CI) | P-value | ES |
| **HDL** | | | | | | | | | | |
| Normal | 1 | | 1 | | | 1 | | 1 | | |
| Low | 0.93(0.38-2.30) | 0.88 | 0.91(0.34-2.38) | 0.84 | -0.05 | 0.83(0.37-1.87) | 0.66 | 0.86(0.38-1.97) | 0.73 | -0.08 |
| **VLDL** | | | | | | | | | | |
| Normal (ref.) | 1 | | 1 | | | 1 | | 1 | | |
| High | 0.88(0.33-2.35) | 0.81 | 0.97(0.76-1.23) | 0.81 | 0.09 | 1.03(0.78-1.37) | 0.79 | 0.92(0.66-1.28) | 0.63 | 0.25 |
| **Cholesterol** | | | | | | | | | | |
| Normal **(ref.)** | 1 | | 1 | | | 1 | | 1 | | |
| High | 1.04(0.90-1.21) | 0.51 | 1.00(0.87-1.16) | 0.92 | 0.19 | 0.98(0.82-1.17) | 0.86 | 0.76(0.61-0.95) | 0.01* | -0.30 |
| **Triglyceride** | | | | | | | | | | |
| Normal **(ref.)** | **1** | | 1 | | | 1 | | 1 | | |
| High | 1.10(0.73-1.66) | 0.59 | 0.93(0.77-1.13) | 0.50 | 0.03 | 0.94(0.77-1.15) | 0.50 | 1.01(0.83-1.23) | 0.88 | 0.17 |
| **CRP** | | | | | | | | | | |
| Normal **(ref.)** | 1 | | 1 | | | 1 | | 1 | | |
| High | 1.08(0.94-1.25) | 0.21 | 1.18(0.90-1.54) | 0.21 | 0.25 | 0.98(0.83-1.15) | 0.85 | 0.96(0.82-1.12) | 0.65 | 0.04 |

**CRP:** C-reactive protein; **BMI:** Body mass index; **FPG:** Fasting plasma glucose; **HDL:** High-density lipoprotein cholesterol; **VLDL:** Very low-density lipoprotein cholesterol; **HOMA-IR:** Homeostatic model assessment of insulin resistance; **HOMA-B:** Homeostatic model of beta-cell function; **WHR:** Weight-to-height ratio; **COVID-19:** Coronavirus disease 2019; **CPR:** Crude prevalence ratio (univariate); **APR:** Adjusted prevalence ratio (multivariate); *: Significant p-value; **ES:** Effect size (Cohen's d)

insulin resistance in our setting (P < 0.05; Table 6). In general, the strength of the observed associations was small. Compared to age group 20–29 years, age 40–49 years demonstrated a 16% increased prevalence of insulin resistance (PR: 1.16; CI: 1.05-1.28; P = 0.003; Table 6) in participants with past COVID-19 status but in participants without past COVID-19 status, 11% increased prevalence of insulin resistance was observed (PR: 1.11; CI: 1.01-1.23; P = 0.03; Table 6) in age group 50–59 years. In the group with past COVID-19 status, participants with high triglycerides level compared with normal level showed a 9% increase in prevalence of insulin resistance (PR: 1.09; CI: 1.03-1.18; P = 0.01; Table 6). However, being underweight (PR: 0.86; CI: 0.77-0.95; P = 0.002; Table 6); overweight (PR: 0.88; CI: 0.78-0.99; P = 0.03; Table 6) or having high level of CRP (PR: 0.88; CI: 0.79-0.99; P = 0.03; Table 6), rather decreased the prevalence of insulin resistance compared with their counterparts with normal levels of these parameters in this group.

## Discussion

Results from our analyses show that irrespective of the surrogate marker of insulin resistance or past COVID-19 status, age, educational level and triglycerides could significantly associate with insulin resistance.

In participants without past COVID-19 status, fasting plasma glucose, insulin and total cholesterol could significantly associate with insulin resistance measured by HOMA-IR. In these participants, high levels of plasma glucose and insulin exhibited 106% and 57% respective increased prevalence of insulin resistance compared with their counterparts with normal levels of the parameters. However, high total cholesterol level showed a 24% reduction in prevalence of insulin resistance. Irrespective of past COVID-19 status and residential location, 74.31% prevalence of insulin resistance was observed with HOMA-IR as the surrogate marker.

Using TyG as the index for insulin resistance, age, BMI, triglycerides and CRP were the significant predictors of insulin resistance in participants with past COVID-19 status. In these participants, being in age 40–49 years compared to age 20–29 years and having high triglycerides level compared to the normal level increased the prevalence of insulin

**Table 6. Predictors of insulin resistance measured by triglyceride-glucose index according to past COVID-19 status.**

| Variable | PAST COVID-19 STATUS | | | | | NO PAST COVID-19 STATUS | | | | |
|---|---|---|---|---|---|---|---|---|---|---|
| | CPR (95%CI) | P-value | APR (95%CI) | P –value | ES | CPR (95%CI) | P-value | APR (95%CI) | P-value | ES |
| **Location** | | | | | | | | | | |
| Cape Coast (**ref.**) | 1 | | 1 | | | 1 | | 1 | | |
| Tamale | 0.93(0.87-1.00) | 0.1 | 0.93(0.85-1.00) | 0.06 | -0.79 | 0.64(0.23-1.76) | 0.49 | 1.00(0.89-1.13) | 0.97 | -0.06 |
| **Sex** | | | | | | | | | | |
| Females (ref.) | 1 | | 1 | | | 1 | | 1 | | |
| Males | 1.01(0.93-1.09) | 0.81 | 1.02(0.94-1.16) | 0.58 | 0.18 | 1.05(0.97-1.13) | 0.22 | 1.07(0.99-1.16) | 0.0.06 | 0.47 |
| **Age** | | | | | | | | | | |
| 20-29 (**ref**) | 1 | | 1 | | | 1 | | 1 | | |
| 30-39 | 1.07(0.97-1.18) | 0.21 | 1.04(0.95-1.16) | 0.38 | 0.39 | 1.05(00.96-1.16) | 0.30 | 1.03(0.94-1.14) | 0.54 | 0.06 |
| 40-49 | 1.15(1.06-1.24) | 0.001 | 1.16(1.05-1.28) | 0.003* | -0.03 | 0.99(0.87-1.14) | 0.94 | 0.98(0.86-1.11) | 0.73 | -0.39 |
| 50-59 | 1.06(0.90-1.26) | 0.46 | 1.09(0.93-1.29) | 0.27 | 0.38 | 1.13(1.05-1.21) | 0.001* | 1.11(1.01-1.23) | 0.03* | -0.01 |
| 60 and above | 0.91(0.70-1.20) | 0.53 | 0.91(0.70-1.19) | 0.49 | 0.21 | 1.05(0.90-1.23) | 0.50 | 0.98(0.85-1.13) | 0.82 | -0.03 |
| **Education** | | | | | | | | | | |
| No education (ref.) | 1 | | 1 | | | 1 | | 1 | | |
| Primary | 0.95(0.78-4.18) | 0.15 | 0.98(0.70-4.00) | 0.11 | -0.01 | 1.02(0.84-1.23) | 0.87 | 1.03(0.85-1.25) | 0.76 | -0.08 |
| Secondary | 0.97(0.69-3.67) | 0.09 | 0.96(0.61-3.11) | 0.06 | -0.02 | 1.12(1.00-1.25) | 0.29 | 1.12(0.99-1.27) | 0.07 | 0.37 |
| Tertiary | 0.99(0.37-1.76) | 0.08 | 0.98(0.49-2.10) | 0.12 | -0.01 | 1.11(0.99-1.244) | 0.06 | 1.11(0.99-1.24) | 0.06 | 0.10 |
| **Smoke** | | | | | | | | | | |
| No (**ref.**) | 1 | | 1 | | | 1 | | 1 | | |
| Yes | 0.95(0.79-1.15) | 0.62 | 0.93(0.76-1.13) | 0.45 | -0.27 | 1.09(1.06-1.14) | <0.001* | 1.08(0.95-1.23) | 0.23 | |
| **HOMA-B** | | | | | | | | | | |
| Normal (ref.) | 1 | | 1 | | | 1 | | 1 | | |
| High | 0.98(0.88-2.13) | 0.12 | 0.87(0.80-1.59) | 0.18 | -0.48 | 0.22(0.10-1.69) | 0.12 | 0.34(0.21-1.20) | 0.20 | -0.86 |
| Low | 0.88(079-1.15) | 0.98 | 0.78(0.67-1.52) | 0.40 | | 0.58(0.20-1.73) | 0.33 | 0.51(0.15-1.64) | 0.28 | -0.39 |
| **WHR** | | | | | | | | | | |
| Normal (**ref.**) | 1 | | 1 | | | 1 | | 1 | | |
| High | 0.88(0.67-1.34) | 0.41 | 0.78(0.20-1.56) | 0.34 | -0.33 | 0.92(0.31-1.75) | 0.34 | 1.32(0.41-2.40) | 0.58 | 0.02 |
| **Waist circumference** | | | | | | | | | | |
| Normal (ref.) | 1 | | 1 | | | 1 | | | | |
| High | 0.99(0.90-1.09) | 0.89 | 0.90(0.78-1.03) | 0.12 | -0.65 | 1.08(0.99-1.16) | 0.05 | 0.96(0.86-1.07) | 0.47 | -0.37 |
| **BMI** | | | | | | | | | | |
| Normal (ref.) | 1 | | 1 | | | 1 | | 1 | | |
| Underweight | 0.87(0.83-0.94) | <0.001* | 0.86(0.77-0.95) | 0.002* | -0.03 | 1.09(0.70-1.70) | 0.69 | 1.15(0.73-1.79) | 0.55 | -0.63 |
| Overweight | 0.95(0.90-0.99) | 0.03* | 0.88(0.78-0.99) | 0.03* | 0.43 | 1.20(0.77-1.87) | 0.41 | 1.28(0.82-1.99) | 0.28 | 0.87 |
| Obese | 0.92(0.81-1.03) | 0.16 | 0.98(0.82-1.17) | 0.80 | 0.34 | 1.25(0.81-1.94) | 0.32 | 1.31(0.83-2.06) | 0.25 | -0.02 |
| **FPG** | | | | | | | | | | |
| Normal (ref.) | 1 | | 1 | | | 1 | | 1 | | |
| High | 1.33(0.76-2.35) | 0.32 | 1.31(0.83-2.05) | 0.24 | 0.75 | 1.19(0.55-2.62) | 0.46 | 1.17(0.57-2.34) | 0.67 | 1.11 |
| **Insulin** | | | | | | | | | | |
| Normal (ref.) | 1 | | 1 | | | 1 | | 1 | | |
| High | 1.26(0.71-2.22) | 0.42 | 1.22(0.78-1.89) | 0.39 | 0.79 | 1.02(0.93-1.11) | 0.72 | 1.00(0.91-1.09) | 0.99 | -0.03 |
| **LDL** | | | | | | | | | | |
| Normal (ref.) | 1 | | 1 | | | 1 | | 1 | | |
| High | 1.02(0.94-1.10) | 0.65 | 0.92(0.85-1.02) | 0.13 | 0.24 | 1.04(0.96-1.12) | 0.39 | 0.95(0.88-1.05) | 0.32 | 0.40 |

*(Continued)*

**Table 6.** (Continued)

| Variable | PAST COVID-19 STATUS | | | | | NO PAST COVID-19 STATUS | | | | |
|---|---|---|---|---|---|---|---|---|---|---|
| | CPR (95%CI) | P-value | APR (95%CI) | P –value | ES | CPR (95%CI) | P-value | APR (95%CI) | P-value | ES |
| **HDL** | | | | | | | | | | |
| Normal (ref.) | 1 | | 1 | | | 1 | | 1 | | |
| Low | 0.78(0.21-2.82) | 0.70 | 0.37(0.09-1.54) | 0.17 | -0.55 | 0.70(0.19-2.56) | 0.59 | 0.83(0.22-3.20) | 0.79 | -0.10 |
| **VLDL** | | | | | | | | | | |
| Normal (ref.) | 1 | | 1 | | | 1 | | 1 | | |
| High | 1.10(1.06-1.15) | <0.001* | 0.99(0.93-1.05) | 0.71 | 0.01 | 0.97(0.82-1.15) | 0.75 | 0.96(0.81-1.13) | 0.62 | -0.54 |
| **Cholesterol** | | | | | | | | | | |
| Normal (ref.) | 1 | | 1 | | | 1 | | 1 | | |
| High | 1.06(0.99-1.14) | 0.09 | 1.02(0.92-1.10) | 0.49 | 0.47 | 1.09(1.02-1.17) | 0.01* | 1.10(0.98-1.24) | 0.08 | 1.01 |
| **Triglyceride** | | | | | | | | | | |
| Normal (ref.) | 1 | | 1 | | | 1 | | 1 | | |
| High | 1.13(1.07-1.18) | <0.001* | 1.09(1.03-1.18) | 0.01* | -0.01 | 1.02(0.93-1.11) | 0.69 | 1.00(0.91-1.10) | 0.63 | -0.01 |
| **CRP** | | | | | | | | | | |
| Normal (ref.) | 1 | | 1 | | | 1 | | 1 | | |
| High | 1.05(1.01-1.11) | 0.03* | 0.88(0.79-0.99) | 0.03* | -0.43 | 0.94(0.87-1.02) | 0.17 | 0.95(0.88-1.03) | 0.28 | -0.42 |

**CRP:** C-reactive protein; **BMI:** Body mass index; **FPG:** Fasting plasma glucose; **HDL:** High-density lipoprotein cholesterol; **VLDL:** Very low-density lipoprotein cholesterol; **HOMA-IR:** Homeostatic model assessment of insulin resistance; **HOMA-B:** Homeostatic model of beta-cell function; **WHR:** Weight-to-height ratio; **COVID-19:** Coronavirus disease 2019; **CPR:** Crude prevalence ratio (univariate); **APR:** Adjusted prevalence ratio (multivariate); **\*:** Significant p-value; **ES:** Effect size (Cohen's d)

resistance. However, being underweight or overweight compared to normal weight and having high level of CRP compared to the normal level decreased the prevalence of insulin resistance. For participants without past COVID-19 status, age group 50–59 years compared with 20–29 years increased the prevalence of insulin resistance as measured by TyG. Irrespective of past COVID-19 status and residential location, TyG-determined insulin resistance prevalence was 89.61%.

Insulin resistance remains a cardinal risk factor for the development of T2DM. The gold standard for the assessment of insulin resistance is the glucose clamp technique developed by DeFronzo et al. [17]. However, the method is expensive, laborious and requires high level of technical expertise, making it unsuitable for application in large-sample studies and routine clinical practice. Therefore, various less expensive and simple techniques that rely on surrogate markers have been developed for effective assessment of insulin resistance. The homeostatic model developed by Matthews et al. [14] to assess insulin resistance (HOMA-IR) and beta-cell function (HOMAB) and the triglyceride-glucose index (TyG) [15] are some of such methods that have been validated against the gold standard and found to be appropriate for assessing insulin resistance under fasting steady state [18,19]. Insulin resistance is affected by race and other environmental factors, which differ from place to place. In the current study, we sought to examine determinants of insulin resistance, assessed by two surrogate markers, in non-diabetic participants with past COVID-19 status compared with their counterparts without any past COVID-19 status in two regional capitals of Ghana. The insulin resistance prevalence of 74.31% and 89.61% as determined by HOMA-IR and TyG respectively in our sample are higher than 38% reported in Lebanon [20], 46.4% in Venezuela [21], and 60.2% in Bangladesh [22]. This extremely high prevalence of insulin resistance exemplifies the risk of development of T2DM in our setting under favourable conditions. Considering that insulin resistance cuts across the various weight groups in our setting, with majority (52.25%) of participants within the normal weight category, other factors beyond being overweight or obese could be accounting for our observation. Indeed, irrespective of the surrogate marker used or past COVID-19 status, the prevalence of insulin resistance could not be predicted by body mass index, waist circumference, waist-to-hip ratio, C-reactive protein, smoking and alcohol intake in our setting although high level

of circulating triglycerides increased the observed prevalence. Our findings appear contrary to those of previous studies [23–29] that associated indices of obesity and inflammation with insulin resistance. However, the observed positive association of insulin resistance with triglycerides level irrespective of past COVID-19 status or surrogate marker used in the current study is in support of studies [25,29] that associated insulin resistance with lipid metabolism in furtherance of the established diabetogenic role of adiposity [30]. Indeed, white adipose tissue has long been implicated in the development of insulin resistance through production of adipocytokines and other lipid derivatives that impair insulin signaling [31–33].

A recent large-sample comprehensive epidemiological study which included 5340 participants and quantified 78 metabolic measures reported that non-diabetic individuals with insulin resistance are exposed to similar adverse metabolic environment in non-fasting state as their diabetic counterparts [34]. This observation implies that the metabolic response of the insulin resistant individual in the fed state may be even more important to the overall health than that of the fasting state used in various diagnostic decisions. Molecular and cellular evidence of IR show that the phenomenon results from altered phosphorylation in the insulin-signaling pathway in different cells and organs with overall adverse effects on carbohydrate, protein and lipid metabolism and associated health risk [35,36]. Therefore, the failure of the above indices of obesity and inflammation to predict insulin resistance assessed under fasting state in our setting irrespective of past COVID-19 status or surrogate marker of insulin resistance does not imply that those biochemical processes are not involved at the molecular level with respect to the development of the condition. The difference in findings between the current study and the previous ones [23–29] could be ascribed to variations in participants characteristics. Indeed, none of these studies [23–29] involved individuals who had recovered from COVID-19. Indeed, sub-group analyses based on past COVID-19 status revealed a seeming protection against insulin resistance prevalence in both underweight and overweight participants with past COVID-19 status only. This finding portrays the complexity of the role of adiposity in the development of insulin resistance in our context.

However, the observation of insulin resistance across the various weight groups corroborates a number of previous studies [20–27] and implies that the development of insulin resistance probably precedes weight gain. Indeed, in spite of variations in mean or median levels of some of the measured biochemical indices between the two study groups, the observed levels were generally within the normal range, further supporting the notion that the development of insulin resistance does not necessarily require high levels of such indices. High levels of such indices may rather worsen an already existing condition. For instance, low-grade inflammation, is known to be associated with insulin resistance [37,38] implying that a high grade version may aggravate the already established condition. Interestingly, the seeming promotion of insulin resistance by high levels of C-reactive protein, which could not be sustained when TyG was the surrogate marker, in the entire sample, irrespective of COVID-19 status, was rather sustained only in the sub-group analysis in participants with past COVID-19 status. This observation, together with the various predictors of insulin resistance identified by the two surrogate markers points to the complexity of the phenomenon and its associated factors.

Interestingly, a seeming protection from HOMA-IR-determined insulin resistance by tertiary education in the entire participants irrespective of past COVID-19 status changed to facilitation with TyG index as the surrogate marker although age 50–59 years was identified as a facilitator of insulin resistance by both surrogate markers in our setting. Moreover, high insulin level could only be found to be an important promoter of HOMA-IR-determined insulin resistance in participants without prior COVID-19 status in our context. Our observations in respect of the association of insulin resistance with high insulin level and advanced age appear to support previous studies [39,40]. High insulin level can be a consequence of insulin resistance but it can also be an independent primary defect indicative of an unfavourable metabolic health [41,42]. High insulin level is influenced by several factors including age, race, genetics, adiposity, free fatty acid levels, insulin secretion and clearance and it is self-perpetuating [41,43]. A cell-based study [44] has long demonstrated that chronic exposure of insulin receptors to high levels of insulin impairs insulin signaling with time through altered autophosphorylation without affecting the quantity of receptors. This finding, which has been corroborated by other studies in humans [41,43] as well as the current one, suggests that once hyperinsulinemic-insulin

resistance cycle is established, metabolic health is adversarially affected if allowed to persist. Although a recent study by Stephens et al. [40] associated metabolic health with age and higher education, the nature of the reported association, notably, for education, differs from that of the current study in the context of TyG-determined insulin resistance. In the current study, tertiary education was positively associated with TyG-determined insulin resistance as opposed to the previous study [40] that reported a negative relationship. However, in terms of HOMA-IR-determined insulin resistance, our findings corroborate those of Stephens et al. [40]. As reported in previous studies [45,46], the attainment of a tertiary education qualification is virtually a pre-requisite for decent employment notably in the formal sector of the Ghanaian economy. Moreover, with the retirement age being 60 years in Ghana, age 50–59 years corresponds with the age range at which participants will likely be at the peak of their career where various incentives that promote job-related sedentariness is prevalent. Such a situation can create a conducive metabolic environment for the development of insulin resistance as observed in the current study.

Interestingly, with TyG index as the surrogate marker, a sub-group analyses showed that in participants with past COVID-19 status, apart from age, formal education and BMI that associated with the observed insulin resistance, high triglycerides level promoted but high CRP level protected against the phenomenon. Whereas the protection offered by high CRP level deviates from the expected facilitation role [37,38], it further demonstrates the complexity of the relationship between CRP and the insulin resistance phenomenon.

Fasting plasma glucose (FPG) and beta-cell function (HOMA-B) are critical factors that are known to be associated with insulin resistance [47–49]. High FPG could be a consequence of insulin resistance or reduced beta-cell function or both although glucose-induced insulin resistance [47,48] is possible, and can create a hyperglycaemia-insulin resistance cycle that requires an intervention to ensure a favourable metabolic health. Therefore, in the current study where insulin resistance is highly prevalent, participants with high fasting plasma glucose are expected to have a heightened risk of developing T2DM. High beta-cell function may demonstrate some protection against development of insulin resistance in the short-term but increases the diabetogenic risk in the long-term due to its probable unsustainable nature through exhaustion [49]. Irrespective of past COVID-19 status, neither FPG nor HOMA-B was associated with insulin resistance in our setting when TyG was used as the index for insulin resistance. However, with HOMA-IR as the surrogate marker, FPG in addition to insulin and total cholesterol levels associated with insulin resistance in participants without past COVID-19 status. This identification of different predictive variables of insulin resistance under different surrogate markers reflects the complexity of the phenomenon and its contributory factors [50].

Our study is not without limitations. Being a cross-sectional study, causal relationships could not be established. Also, detailed information on other lifestyle factors such as exercise and dietary information which can affect insulin resistance were not available. Above all, the use of fasting samples that are subject to fluctuations call for caution in undue extrapolation of our findings to populations of different characteristics. Therefore, longitudinal studies with wider coverage and larger sample size involving additional biomarkers that assess both fasting and postprandial insulin and glucose indices may be necessary to better understand the promoting factors of insulin resistance in our context. In spite of the above limitations, the current study has provided evidence of factors that affect the development of insulin resistance in participants with and without past COVID-19 status in our setting.

## Conclusion

The extremely high prevalence of insulin resistance in both groups of participants in addition to dyslipidemias and abnormal beta-cell function connotes increased diabetogenic risk. Irrespective of surrogate marker or past COVID-19 status, age, education and triglycerides levels associated with insulin resistance. In participants without prior COVID-19 status, FPG, insulin and total cholesterol associated with HOMA-IR-determined insulin resistance. However, with TyG index, age, BMI, triglycerides and CRP were the significant determinants of insulin resistance in participants with past COVID-19 status. The use of HOMA-IR and TyG as surrogate markers for insulin resistance

allowed for the identification of a wide range of predictors of the condition in our context, which could not have been possible if only one surrogate marker had been used. Further studies are needed to unravel additional factors that influence the development of insulin resistance under fasting and postprandial states in apparently healthy adults in our setting even as health educational measures on lifestyle changes that improve insulin sensitivity are intensified.

## Supporting information

**S1 Checklist.  Inclusivity in global research.**
(DOCX)

## Acknowledgments

The authors are very grateful to the study participants for their unique contribution in ensuring the success of this research work. Additionally, we are grateful to the staff and management of the Cape Coast and Tamale Teaching Hospitals for allowing their facilities to be used as sites for the study. Above all, we are thankful to the staff and management of the Public Health Reference Laboratory of the Tamale Teaching Hospital for their support in the conduct of the study.

## Author contributions

**Conceptualization:** Ansumana Sandy Bockarie, Leonard Derkyi-Kwarteng, Samuel Acquah.

**Data curation:** Ansumana Sandy Bockarie, Leonard Derkyi-Kwarteng, Jeffrey Amankona Obeng, Richard Kujo Adatsi, Ebenezer Aniakwaa-Bonsu, Charles Apprey, Jerry Ampofo-Asiama, Samuel Acquah.

**Formal analysis:** Ansumana Sandy Bockarie, Leonard Derkyi-Kwarteng, Jeffrey Amankona Obeng, Richard Kujo Adatsi, Ebenezer Aniakwaa-Bonsu, Charles Apprey, Jerry Ampofo-Asiama, Samuel Acquah.

**Funding acquisition:** Jeffrey Amankona Obeng, Richard Kujo Adatsi, Samuel Acquah.

**Investigation:** Jeffrey Amankona Obeng, Richard Kujo Adatsi, Samuel Acquah.

**Methodology:** Ansumana Sandy Bockarie, Leonard Derkyi-Kwarteng, Jeffrey Amankona Obeng, Richard Kujo Adatsi, Ebenezer Aniakwaa-Bonsu, Charles Apprey, Jerry Ampofo-Asiama, Samuel Acquah.

**Project administration:** Ansumana Sandy Bockarie, Jeffrey Amankona Obeng, Richard Kujo Adatsi, Charles Apprey, Samuel Acquah.

**Resources:** Jeffrey Amankona Obeng, Richard Kujo Adatsi, Samuel Acquah.

**Software:** Jeffrey Amankona Obeng, Richard Kujo Adatsi, Ebenezer Aniakwaa-Bonsu, Charles Apprey, Jerry Ampofo-Asiama, Samuel Acquah.

**Supervision:** Ansumana Sandy Bockarie, Leonard Derkyi-Kwarteng, Samuel Acquah.

**Validation:** Leonard Derkyi-Kwarteng, Jeffrey Amankona Obeng, Richard Kujo Adatsi, Ebenezer Aniakwaa-Bonsu, Charles Apprey, Jerry Ampofo-Asiama, Samuel Acquah.

**Visualization:** Leonard Derkyi-Kwarteng, Jeffrey Amankona Obeng, Richard Kujo Adatsi, Ebenezer Aniakwaa-Bonsu, Charles Apprey, Jerry Ampofo-Asiama, Samuel Acquah.

**Writing – original draft:** Ansumana Sandy Bockarie, Leonard Derkyi-Kwarteng, Ebenezer Aniakwaa-Bonsu, Charles Apprey, Jerry Ampofo-Asiama, Samuel Acquah.

**Writing – review & editing:** Ansumana Sandy Bockarie, Leonard Derkyi-Kwarteng, Charles Apprey, Jerry Ampofo-Asiama, Samuel Acquah.

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
