## [Decision Letter · Decision Letter 0]

10 Jun 2024

PGPH-D-24-00529

PREVALENCE AND DETERMINANTS OF INSULIN RESISTANCE IN RESIDENTS OF TWO REGIONAL CAPITALS IN GHANA: AN OBSERVATIONAL STUDY

Dear Dr.Samuel Acquah

Thank you for submitting your manuscript to PLOS Global Public Health. After careful consideration, we feel that it has merit but does not fully meet PLOS Global Public Health’s publication criteria as it currently stands. Therefore, we invite you to submit a revised version of the manuscript that addresses the points raised during the review process.

We look forward to receiving your revised manuscript.

Kind regards,

Nitu Nigam, PhD Medical Genetics

Academic Editor

Journal Requirements:

2. We ask that a manuscript source file is provided at Revision. Please upload your manuscript file as a .doc, .docx, .rtf or .tex.

Additional Editor Comments (if provided):

Reviewers' comments:

Reviewer's Responses to Questions

**Comments to the Author**

1. Does this manuscript meet PLOS Global Public Health’s publication criteria ? Is the manuscript technically sound, and do the data support the conclusions? The manuscript must describe methodologically and ethically rigorous research with conclusions that are appropriately drawn based on the data presented.

Reviewer #1: Yes

2. Has the statistical analysis been performed appropriately and rigorously?

Reviewer #1: Yes

3. Have the authors made all data underlying the findings in their manuscript fully available (please refer to the Data Availability Statement at the start of the manuscript PDF file)?

Reviewer #1: Yes

4. Is the manuscript presented in an intelligible fashion and written in standard English?

Reviewer #1: Yes

5. Review Comments to the Author

Reviewer #1: While the article provides valuable insights into the potential long-term health impacts of COVID-19, particularly regarding the development of type 2 diabetes mellitus (T2DM) and insulin resistance in the sub-Saharan African population, it has several limitations:

1. The use of convenience sampling and recruitment from only two healthcare facilities could introduce selection bias, as participants may not be representative of the general population. This could limit the external validity of the study.

2. While the study aims to identify determinants of insulin resistance in individuals with a history of COVID-19 compared to those without, it does not explore potential mechanisms underlying the relationship between COVID-19 and insulin resistance.

3. The study's use of fasting samples may not capture postprandial changes in insulin sensitivity, which could affect the interpretation of results.

4. The study lacks detailed information on lifestyle factors such as diet and exercise, which are known to influence insulin resistance.

5. The study's reliance on cross-sectional data limits its ability to track changes in insulin resistance over time. Longitudinal studies would provide more robust evidence of the long-term effects of COVID-19 on insulin sensitivity.

6. While the study conducts statistical analyses to assess associations between variables, it does not explicitly address the possibility of type I or type II errors, which could affect the reliability of study findings. Additionally, the interpretation of statistical significance should consider clinical relevance and effect sizes.

6. PLOS authors have the option to publish the peer review history of their article (what does this mean? ). If published, this will include your full peer review and any attached files.

**Do you want your identity to be public for this peer review?** For information about this choice, including consent withdrawal, please see our Privacy Policy .

Reviewer #1: **Yes: ** Profm Mohd Ashraf Ganie

---

## [Editor Report · Decision Letter 1]

23 Sep 2024

PGPH-D-24-00529R1

PREVALENCE AND DETERMINANTS OF INSULIN RESISTANCE IN RESIDENTS OF TWO REGIONAL CAPITALS IN GHANA: AN OBSERVATIONAL STUDY

Dear Dr. Acquah,

Thank you for submitting your manuscript to PLOS Global Public Health. After careful consideration, we feel that it has merit but does not fully meet PLOS Global Public Health’s publication criteria as it currently stands. Therefore, we invite you to submit a revised version of the manuscript that addresses the points raised during the review process.

We look forward to receiving your revised manuscript.

Kind regards,

Paraskevi Detopoulou

Academic Editor

Journal Requirements:

Additional Editor Comments (if provided):

Several comments should be addressed. A general comment if that the results section is a bit confusing since several ways of ins resistance are considered. Results should be clear-cut (maybe choose one model).

The second paragraph of the introduction should be shorter (by half).

Insulin resistance, which signifies reduced sensitivity of cells to insulin cues, is a cardinal risk factor for the development of T2DM. Insulin resistance is influenced by several factors including infection, obesity, diet and other lifestyle factors. These sentences can be deleted.

Paragraph: Demographic, Blood Pressure, Anthropometry and Biochemical Indices

An experience nurse measured. Please change to “An experienced nurse measured”

Table 1: It is not very clear which comparisons are made for the p-values displayed. Please provide an explanation under the Table.

Table 2 and throughout the text: HDL, LDL and VLDL should be : HDL-cholesterol, LDL-cholesterol and VLDL- cholesterol.

It is not clear of one or several logistic regression models were applied. It is suggested to provide data on simple and more complex models. Moreover, some indices may not be used together in the same model (for example waist circ and WHR). It is strongly suggested to apply models with multiple explanatory variables at once to see if the effects of each variable are lessened in the presence of other variables.

Table 5: ; SE: Effect size do you mean: ES: Effect size

It would be also interesting to have data on COVID vaccination.

The logistic regression models of both ways estimation of insulin resistance is confusing. You can choose one way and the other one can be moved to supplementary data.

How do authors explain that living in Tamala was a protective factor against insulin resistance?

Do we have any data regarding supplements intake? It is known that nutrition plays a role in COVID-19 by modulating several pathways of inflammation including PAF (https://www.ncbi.nlm.nih.gov/pmc/articles/PMC7911163/).
---

## [Decision Letter · Decision Letter 2]

4 Feb 2025

PGPH-D-24-00529R2

PREVALENCE AND DETERMINANTS OF INSULIN RESISTANCE IN RESIDENTS OF TWO REGIONAL CAPITALS IN GHANA: AN OBSERVATIONAL STUDY

Dear Dr. Acquah,

Thank you for submitting your manuscript to PLOS Global Public Health. After careful consideration, we feel that it has merit but does not fully meet PLOS Global Public Health’s publication criteria as it currently stands. Therefore, we invite you to submit a revised version of the manuscript that addresses the points raised during the review process.

We look forward to receiving your revised manuscript.

Kind regards,

Giridhara Rathnaiah Babu, MBBS, MPH, PhD

Academic Editor

Additional Editor Comments (if provided):

I agree with reviewer comments. Please address the corrections suggested by reviewers in introduction, methods, results and discussion.

Please provide details for the rationale for using COVID19 exposure status and the apparent contradiction in methodology (hospital based) and the description that it is community based.

Also, Predictors of insulin resistance is well document in literature, the authors need to explain the novelty of the study.

Reviewers' comments:

Reviewer's Responses to Questions

**Comments to the Author**

1. If the authors have adequately addressed your comments raised in a previous round of review and you feel that this manuscript is now acceptable for publication, you may indicate that here to bypass the “Comments to the Author” section, enter your conflict of interest statement in the “Confidential to Editor” section, and submit your "Accept" recommendation.

Reviewer #2: (No Response)

Reviewer #3: (No Response)

2. Does this manuscript meet PLOS Global Public Health’s publication criteria ? Is the manuscript technically sound, and do the data support the conclusions? The manuscript must describe methodologically and ethically rigorous research with conclusions that are appropriately drawn based on the data presented.

Reviewer #2: Partly

Reviewer #3: Partly

3. Has the statistical analysis been performed appropriately and rigorously?

Reviewer #2: Yes

Reviewer #3: Yes

4. Have the authors made all data underlying the findings in their manuscript fully available (please refer to the Data Availability Statement at the start of the manuscript PDF file)?

Reviewer #2: Yes

Reviewer #3: Yes

5. Is the manuscript presented in an intelligible fashion and written in standard English?

Reviewer #2: Yes

Reviewer #3: Yes

6. Review Comments to the Author

Reviewer #2: I have only one comment to the authors. I think the title does not fully capture the focus of the study. The current title, "Prevalence and Determinants of Insulin Resistance in Residents of Two Regional Capitals in Ghana: An Observational Study", provides some information, and it does not highlight the aspect of COVID-19 exposure as one of the determinants which has been widely explored in the study. A revised tittle should reflect the study's objectives and design.

Reviewer #3: (No Response)

7. PLOS authors have the option to publish the peer review history of their article (what does this mean? ). If published, this will include your full peer review and any attached files.

**Do you want your identity to be public for this peer review?** For information about this choice, including consent withdrawal, please see our Privacy Policy .

Reviewer #2: No

Reviewer #3: **Yes: ** David Lubogo

---

## [Decision Letter · Decision Letter 3]

24 Mar 2025

PREVALENCE AND DETERMINANTS OF INSULIN RESISTANCE IN RESIDENTS OF TWO REGIONAL CAPITALS IN GHANA: AN OBSERVATIONAL STUDY

PGPH-D-24-00529R3

Dear Asssociate Professor Acquah,

We are pleased to inform you that your manuscript 'PREVALENCE AND DETERMINANTS OF INSULIN RESISTANCE IN RESIDENTS OF TWO REGIONAL CAPITALS IN GHANA: AN OBSERVATIONAL STUDY' has been provisionally accepted for publication in PLOS Global Public Health.

Best regards,

Giridhara Rathnaiah Babu, MBBS, MPH, PhD

Academic Editor

Reviewer Comments (if any, and for reference):

Reviewer's Responses to Questions

**Comments to the Author**

1. If the authors have adequately addressed your comments raised in a previous round of review and you feel that this manuscript is now acceptable for publication, you may indicate that here to bypass the “Comments to the Author” section, enter your conflict of interest statement in the “Confidential to Editor” section, and submit your "Accept" recommendation.

Reviewer #2: All comments have been addressed

Reviewer #3: All comments have been addressed

2. Does this manuscript meet PLOS Global Public Health’s publication criteria ? Is the manuscript technically sound, and do the data support the conclusions? The manuscript must describe methodologically and ethically rigorous research with conclusions that are appropriately drawn based on the data presented.

Reviewer #2: Yes

Reviewer #3: Partly

3. Has the statistical analysis been performed appropriately and rigorously?

Reviewer #2: Yes

Reviewer #3: I don't know

4. Have the authors made all data underlying the findings in their manuscript fully available (please refer to the Data Availability Statement at the start of the manuscript PDF file)?

Reviewer #2: Yes

Reviewer #3: Yes

5. Is the manuscript presented in an intelligible fashion and written in standard English?

Reviewer #2: Yes

Reviewer #3: Yes

6. Review Comments to the Author

Reviewer #2: I do not have any further comments. I recommend this article for publication.

Reviewer #3: I thank the authors for improving this manuscript. Most of the issues have been addressed.

I have indicated some few comments for the authors to consider in the manuscript.

Thank you.

7. PLOS authors have the option to publish the peer review history of their article (what does this mean? ). If published, this will include your full peer review and any attached files.

**Do you want your identity to be public for this peer review?** For information about this choice, including consent withdrawal, please see our Privacy Policy .

Reviewer #2: No

Reviewer #3: **Yes: ** David Lubogo
